# Pull-apart basin tectonic model is structurally impossible for Kashmir basin, NW Himalaya

A. A. Shah

Physical & Geological Sciences, Faculty of Science Universiti Brunei Darussalam, Brueni

Correspondence to: A. A. Shah (afroz.shah@gmail.com)

**Abstract:** Kashmir Basin in NW Himalaya is considered a Neogene-Quatermary piggyback basin that was formed as result of the continent-continent collision of Indian and Eurasian plates. This model however is recently challenged by a pull-apart

basin model, which argues that a major dextral strike-slip fault through Kashmir basin is responsible for its formation. And here it is demonstrated that the new tectonic model is structurally problematic, and conflicts with the geomorphology, geology, and tectonic setting of Kashmir basin. It also conflicts, and contradicts with the various structural features associated with a typical dextral strike-slip fault system where it shows that such a major structure cannot pass through the middle of the basin. It is demonstrated that such a structure is structurally, and kinematically impossible, and could not exist.

#### **1** Introduction

Kashmir basin of NW Himalaya is a typical example of a piggyback basin that was forced as a result of the continent-continent collision of Indian and Eurasian plates (Burbank and Johnson, 1982). This tectonic model has been recently challenged by Alam et al. (2015, 2016). They have introduced a pull-apart basin model to argue that Kashmir basin was formed as a result of a large dextral-strike-slip fault that runs through it. This model however is structurally impossible (Shah, 2015b), and the present work further shows why Kashmir basin is not a pull-apart basin, and demonstrates that such a setting is inconsistent

with the geology, topographic, geomorphology, and structural architecture of the basin. It however supports the classical piggyback basin model for its formation (Burbank and Johnson, 1983), and demonstrates that active movement on the NE dipping Kashmir basin fault (KBF) has greatly influenced the geomorphology of the basin.

#### 25 2 Tectonic Background

Kashmir Basin is located on the NW portion of the active continent-continent collision of Indian and Eurasian plates (Burbank and Johnson, 1982). The oval shaped basin filled with young sediments is considered a classical example of the piggyback basin of Neogene-Quatermary intermontane basins of the Himalayas (Burbank and Johnson, 1982). The bedrock geology suggests that the Upper Carboniferous-Permian Panjal Volcanic Series and Triassic limestone are the foundation rocks on

which the Plio-Pleistocene fluvio-glacial sediments are deposited (Farooqi and Desai 1974). These sediments constitute a

1,300-m thick sequence of unconsolidated clays, sands, and conglomerates with lignite beds unconformably lying on the bedrock with a cover of recent river alluvium (Bhatt, 1976; Burbank and Johnson 1982; Singh 1982). This whole sequence is cut by a number of active thrust faults (Shah, 2013a, 2013b, 2015a; Ahmad et al., 2013; Shabir and Bhat, 2012), which are scattered through the oval shaped Kashmir basin (KB) that has a strike length of ~150 km and is ~50 km across (Fig. 1). The recent deformation, along these faults, has modified the regional topography, which suggests that the regions on the NE of the

5 Kashmir Basin are subsided, while those on the SW are uplifted (Shah, 2013a). The geomorphic expression of the active thrust fault trace is visible for a distance of ~120 km (Shah, 2013a), wherein it cuts through ~50,000-100,000 year old sediments (e.g. Jaiswal et al., 2009), however, these dates do not show the dated sample locations on a map, which makes it harder to correlate these numbers with the faulting events.

#### 10 3 Pull-apart basin model impossible for Kashmir basin

#### **3.1 Structural evidence**

The strike-length of Kashmir basin is ~150 km, and the mapped length of dextral strike-slip fault is ~165 km, and it runs though the center of the basin, which is structurally impossible (Fig. 2). If a major strike-slip fault has produced a pull-apart basin then the trace of that fault ought not to run in the middle of the basin. And to form ~165 km long basin a series of ~SW,

and ~NE dipping normal faults are required (Fig. 2d). No evidence of such structures are preserved in Kashmir basin. This 15 will also have a unique skeleton that could be related to topography, and geomorphology observed in an area with oppositely dipping normal faults, and that is not seen anywhere in Kashmir basin.

Further, the strike-length of the major dextral-strike slip fault is ~planar, and contiguous, and such geometry cannot cause extension to form a pull-apart basin, and in contrary such basins are typical features of step-overs and linkage fault geometries

20 (Fig. 2).

> And typically horsetail splay faults curve from the main truck of the fault trace (Sylvester, 1988; Burg and Ford, 1997), and in a classic dextral strike-slip fault system such faults could be of certain restricted orientation with respect to the main fault (Figs. 2 and 3). The orientation of the major strike-slip fault of Kashmir basin is reported to be ~NW-SE (Alam et al., 2015), and the horsetail faults are shown to be of the same orientation as the major fault (~NW-SE), this is structurally impossible

(Fig. 3), and conflicts with the basin style of such faulting. Because with ~NW-SE strike of the major fault the horsetail splay 25 structures will have either SW strike with a NW tectonic transport, or NE strike with a SE tectonic transport (Fig. 3c, d).

#### 3.2 Geologic and geomorphic evidence

The bedrock geology of Kashmir basin shows Upper Carboniferous-Permian Panjal Volcanic Series and Triassic limestone are covered by Plio-Pleistocene fluvio-glacial sediments (Farooqi and Desai 1974). There is no evidence of a large scale topographic, or lithology offset, which is typically associated with a major dextral strike-slip fault system. Shah (2013 b)

showed dextral offset of drainages on the SE of Kashmir basin, however, minor ( $\sim 20$  to  $\sim 40$  m) offset of these channels are related to the oblique convergence. And does not suggest or approve of a major dextral strike slip fault system.

The present geological, and structural architecture is consistent major ~NE dipping thrust faults (Fig. 1). The regional structural architecture is consistent with a piggy-back model, where Kashmir basin is riding on a number of thrust faults. And three (Fig.

- 4) SSW verging faults are presently considered active (e.g. Thakur, 2010; Shabir and Bhat 2012; Vasilios 2014; Shah, 2013, 2015), from south these are Main Frontal Thrust (MFT), Medlicott-Wadie Thrust (MWT), and Kashmir Basin Fault (KBF). And the current geomorphic, and structural setup of Kashmir basin is a product of the activity along these faults. It is impossible to create the present structural skeleton of Kashmir basin by a major dextral strike-slip fault, even if it has an oblique slip component (Figs. 3 and 4). This is because if a major dextral-slip is associated with a normal dip-slip component, which is
- shown by the pull-apart model (Alam et al., 2015) then the overall topography, and geomorphology will suggest subsidence on hanging-wall portions, and relative uplift on foot-wall. This requires two scenarios; a) the major fault may either be dipping SSW or 2) NNE. The pull-apart model (Ahmad et al., 2015) shows topographic depression on the left side of the major fault (NNE side), which requires a NNE dipping fault with a normal faulting component (Fig. 3a). However, the entire Kashmir basin tilts ~NE (Fig. 1 and 3), and there is no evidence of normal faulting. There is no reported break or offset in topographic
- with an enormous amount of slip, which could be associated with a dextral-fault. Neither is there any evidence of a large scale strike-slip displacement of bedrock units (Burbank and Johnson, 1983).

#### 3.3 Geodetic evidence

Shah (2013b) mapped the eastern extent of the KBF fault, and argued for a clear right-lateral strike-slip motion for a distance of  $\sim$ 1km, which was shown by the deflection of young stream channels. The lateral offset varies from  $\sim$ 20 to  $\sim$ 40 m. This was

- suggested to be a classical example of oblique convergence where thrusting is associated with a small component of dextral strike-slip motion. The recently acquired GPS data in Kashmir Himalaya (Schiffman et al., 2013a) confirms these observations, and suggests an oblique faulting pattern, wherein a range-normal convergence of 11±1 mm/y is associated with a dextral-shear slip of 5±1 mm/y. They further suggest that obliquity is more towards the eastern portion of the valley. This clearly suggests that the resultant stress vector is oblique in Kashmir Himalaya, and thus the deformation is mainly absorbed by range-normal, and less so by shear components. A typical characteristic feature of oblique convergence
- and less so by shear components. A typical characteristic feature of oblique convergence.

#### 4 Discussion

The Kashmir basin is suggested to have formed through a piggyback basin deformation style, however new studies (Alam et al., 2015, 2016) show it as a large pull-apart basin. The new model is structurally impossible, and conflicts with the geomorphology, geology, and tectonic setting of Kashmir basin. Thus, the possibility of having a major dextral-slip fault in

30 Kashmir basin is kinematically impossible, and it basically conflicts with the typical structural features that are possible in such an environment (Fig. 3). For example horsetail thrust structures do not run parallel with the trend of the main fault trace (Fig. 3), and it is kinematically impossible to have them on both the sides of a major fault tip (Fig. 3). It is equally impossible

to have the trace of a major strike-slip fault in the middle of a pull-apart basin (Fig. 2). Such a structural setting is inconsistent, and impossible. The structural architecture, and the evidence presented above suggests that Kashmir basin does not require a major strike-slip fault. And the conflicting structures that have been shown in pull-apart model are an indication that such a big-structure is not possible in Kashmir basin. Thus, the geological, and tectonic setting of Kashmir basin is largely consistent with a piggy-back model (Burbank and Johnson, 1982). It does not support a large dextral fault, which is structurally impossible

- (see above), and contradicts with the basic geological, and structural set-up of Kashmir basin. The location of the basin is north of the Main Frontal Thrust (MFT), the megathrust structure that accommodates ~2 cm/year of a total of 4-5 cm/year of the regional convergence between the Indian and Eurasian plates (Schiffman et al., 2013; Vasilios 2014), and is considered actively growing (e.g. Bollinger et al., 2014; Malik et al., 2014; Kumahara & Jayangondaperumal,
- 2013; Malik et al., 2010). And the surficial trace of the MFT has not been reported in Jammu and Kashmir, and thus it is assumed that it runs as a blind structure under Jammu (Vassallo et al., 2015) This fault is presently locked under the Kashmir region, and has the potential to host a major earthquake (Schiffman et al., 2013). There is another major active fault that runs under Raisi (Fig.1), which is also considered to host a major earthquake (Vassallo et al., 2015). The third major fault runs approximately through the middle of the Kashmir valley (Fig. 2b) and it can also host a major earthquake, very similar to the
- Muzaferabad earthquake of 2005 (Shah, 2013a). The three active major faults that are mapped in Jammu and Kashmir are not well studied, and there are greater uncertainties in understanding whether the regional stress will be released on the frontal fault or on the interior faults. All these faults are ~SW verging, and can be easily explained by the regional stress setup that a piggyback basin model portrays. However, pull-apart setting (Alam et al., 2015) requires the regional stress to be partitioned into normal, and shear components, which is not shown by seismicity, topography, and structural setting (see above).
- Moreover the basic conceptual conditions for a major dextral strike-slip system are lacking (see above), thus, such a model is impossible in Kashmir basin.

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
