# Peer review of "Pull-apart basin tectonic model is structurally impossible for Kashmir basin, NW Himalaya"

_Solid Earth, 2016_

## Referee Comment (RC1) · Anonymous Referee #1 · 9 Feb 2016

This paper almost reads like a personal diatribe. The author is adamant that the Kashmir Basin is not a pull-apart basin as proposed by Alam et al. (2015, 2016) and the paper is essentially an earnest attempt at refutation. The author calls the pull-apart model 'impossible' 15 times (including in the title and in 110 lines of text) and also states that the pull-apart architecture 'could not exist', is 'problematic' and 'inconsistent with data'. If one of my undergraduates had written this paper, I would have sent it back with advice to remove the redundancy, improve the English, remove absolute terms like 'impossible', eliminate the undercurrents of emotion, and just stick to data-based arguments. This paper is poorly written and should not be published as is.

Now for the science:

1) The author has a limited understanding of oblique deformation and thinks a pull-

apart basin has to have the architecture of the simple cartoon shown in his figure 2. The same holds true for his model-driven views of horsetail splay termination zones (his fig. 3). Transtensional and transpressional fault networks can be highly variable as documented all over the world. Positive and negative flower structures can have a wide variety of fault patterns. Transtensional flower structures do not have to have pull-apart geometries. The authors architectural arguments against a transtensional basin are weak.

2) Neither Shah nor Alam present focal mechanism solutions for earthquakes from the Kashmir Basin. If earthquake focal mechanism solutions revealed any transtensional or extensional events, then the Alam et al. pull-apart models would be more convincing. I have not been able to find any transtensional earthquake events in the Kashmir Basin from my web trawl of relevant literature.

3) Both Alam et al. and Shah should look more carefully at the GPS data in Schiffman et al. (2013). Figure 13 in Alam et al. (2015) is an inaccurate representation of the actual data. It should never have been published. Some of the arrows in their figure are incorrectly oriented and the vector lengths are all the same which is misleading. The Schiffman et al. (2013) GPS data indicate that south-directed motions in Zanskar are oblique to the NW-striking Balapora Fault and Central Kashmir Fault. The obliquity suggests significant components of dextral slip. GPS velocities in Zanskar have higher S and SW velocity components than the data from the Pir Panjal Range. Thus the boundary in between - the CKF – is also under compression. Therefore, the GPS data from Schiffman et al. (2013) suggest dextral transpression within the Kashmir Basin, not transtension. Neither author raises this point correctly, nor mentions the term transpression at all.

4) Shah should point out the unconvincing images of strike-slip related features in the Alam et al. (2015) paper – e.g., their figure 8. I am not convinced of any of their visual 'signatures' of strike slip features and visual offsets.
[Figure]

5) In conclusion, Shah is correct to question the evidence for a pull-apart model for the Kashmir Basin. It does not appear well supported. But he should use concise scientific arguments focusing on the evidence form the historical earthquake record, a more careful analysis of the GPS data base, a less rigid approach to what a transtensional basin has to look like in terms of fault geometries, and a more rigorous scrutiny of the tenuous lineament and apparent offset evidence presented in the Alam et al. papers.

---

## Author Comment (AC1) · 10 Feb 2016

Reply: Dear editor, and the reviewer: Thanks for your time in reviewing my work. I am very pleased to read the comments on my small contribution.

The comments are answered below:

Comment:

Anonymous Referee #1 This paper almost reads like a personal diatribe.

Reply:

I have not written it to attack my colleagues but to discuss science.
[Figure]

Comment:

The author is adamant that the Kashmir Basin is not a pull-apart basin as proposed by Alam et al. (2015, 2016) and the paper is essentially an earnest attempt at refutation. The author calls the pull-apart model 'impossible' 15 times (including in the title and in 110 lines of text) and also states that the pull-apart architecture 'could not exist', is 'problematic' and 'inconsistent with data'. If one of my undergraduates had written this paper, I would have sent it back with advice to remove the redundancy, improve the English, remove absolute terms like 'impossible', eliminate the undercurrents of emotion, and just stick to data-based arguments. This paper is poorly written and should not be published as is.

Reply:

I am sorry if you felt that I am forcing the reader to believe me. It is a language problem, and I am very thankful that you highlighted it. And thanks for your advice, and suggestion. I am here to learn. Modified the entire manuscript as suggested, tried my best.

Comment:

Now for the science: 1) The author has a limited understanding of oblique deformation and thinks a pull apart basin has to have the architecture of the simple cartoon shown in his figure 2. The same holds true for his model-driven views of horsetail splay termination zones (his fig. 3).

Reply:

Surely, I am a new researcher, and I always feel I am a student. I might have, what you call, limited understanding of oblique deformation but I don't think that my figure 2 represents the pull-apart settings around the world. It primarily aims to show the major problem in Alam et al. 2014. They are arguing that Kashmir basin is a pull-apart basin but the major dextral strike-slip fault that they show runs through the center of

the Kashmir basin, which is a basic concern. And the structural orientation of horsetail structures is impossible. When I say impossible it means that such structures cannot form with that orientation (strike, and dip direction of horse tails). This is a major concern, and the evidence that they provide are very weak, and controversial, thus, based on the evidence the authors have produced it is clear that a major dextral fault in Kashmir basin is unconvincing, and wrong.

Since you have questioned my understanding about the figures 2 and 3 could you please scientifically show me how one can get those structures in a proposal dextral-pull apart model. May be I am wrongly thinking. I would greatly appreciate it. In fact you can publish your comment, the journal allows that.

Comment:

Transtensional and transpressional fault networks can be highly variable as documented all over the world. Positive and negative flower structures can have a wide variety of fault patterns. Transtensional flower structures do not have to have pull-apart geometries. The authors architectural arguments against a transtensional basin are weak.

Reply:

According to Burg, J.P "Transpression means that shortening is taking place across a dominantly strike-slip fault (oblique convergence, like along the San Andreas Fault Zone). Conversely, transtension means that extension is a deformation component of bulk strike-slip faulting (California Gulf).". Fossan and Tikoff define it as "Transpression and transtension are broadly defined as steep strike-slip influenced deformation zones that deviate from simple shear by a component of shortening (transpression) or extension (transtension) across the zone".

Fossen, H. and Tikoff, B., 1998. Extended models of transpression and transtension, and application to tectonic settings. Geological Society, London, Special Publications,

135(1), pp.15-33.

Kashmir basin doesn't indicate any kind of transpression, as there is no evidence of a large dextral faulting. There is a slight oblique dextral component along with the dominant thrusting, which is very normal in an oblique convergence environment with a small component of shearing.

The recently acquired GPS data in Kashmir Himalaya (Schiffman et al., 2013) confirms these observations, and suggests an oblique faulting pattern, wherein a range-normal convergence of $11\pm1$ mm/y is associated with a dextral-shear slip of $5\pm1$ mm/y. They further suggest that obliquity is more towards the eastern portion of the valley. This clearly suggests that the resultant stress vector is oblique in Kashmir Himalaya, and thus the deformation is mainly absorbed by range-normal, and less so by shear components. A typical characteristic feature of oblique convergence. Fig. 1b. attached below.

Comment:

2) Neither Shah nor Alam present focal mechanism solutions for earthquakes from the Kashmir Basin. If earthquake focal mechanism solutions revealed any transtensional or extensional events, then the Alam et al. pull-apart models would be more convincing. I have not been able to find any transtensional earthquake events in the Kashmir Basin from my web trawl of relevant literature.

Reply:

I have produced focal mechanism solutions in 2013 paper (Shah, 2013): Shah, A.A.: Earthquake geology of the Kashmir Basin and its implication for large earthquakes. Int. J. Earth. Sci, 102, 7, 1957-1966, 2013. These dominantly show thrust faulting (Fig. 1a). However, the data is limited as the earthquakes in and around Kashmir basin are small, and thus focal mechanism solutions are not available. How can pull-apart model become convincing when their model fails the basic definition of a dextral tectonics.

If we start with a simple pull-apart model then the major trace of the dextral fault that forms the basin should be ~ bordering the basin, and ought not to be through the basin? Isn't that true? That is my major concern, other things are details.

Comment:

3) Both Alam et al. and Shah should look more carefully at the GPS data in Schiffman et al. (2013). Figure 13 in Alam et al. (2015) is an inaccurate representation of the actual data. It should never have been published. Some of the arrows in their figure are incorrectly oriented and the vector lengths are all the same which is misleading. The Schiffman et al. (2013) GPS data indicate that south-directed motions in Zanskar are oblique to the NW-striking Balapora Fault and Central Kashmir Fault. The obliquity suggests significant components of dextral slip. GPS velocities in Zanskar have higher S and SW velocity components than the data from the Pir Panjal Range. Thus the boundary in between - the CKF – is also under compression. Therefore, the GPS data from Schiffman et al. (2013) suggest dextral transpression within the Kashmir Basin, not transtension. Neither author raises this point correctly, nor mentions the term transpression at all.

Reply:

The actual figure from Schiffman et al., 2013 is attached (Fig. 1b), and they have concluded that "GPS measurements in Kashmir Himalaya reveal range normal convergence of 11 ± 1 mm/yr with dextral shear of 5 ± 1 mm/yr." This surely is not transpression? And that is why Schiffman et al, did not use the work transpression in their contribution. The GPS points near Kashmir basin dominantly show oblique convergence with a large component of thrusting. And if we imagine that Kashmir Central Fault exists then the GPS points show dominantly normal convergence not dominant shearing on it (see Fig. 2).

The reason we get more dextral slip towards SE of Kashmir basin is possibly because of the regional escape tectonics, where India acts like an indenter, and the crustal flow
is mostly along the huge strike-slip faults ((Tapponier and Molnar, 1976).Tapponnier, P. and Molnar, P., 1976. Slip-line field theory and large-scale continental tectonics. Nature, 264(5584), pp.319-324.

Comment:

4) Shah should point out the unconvincing images of strike-slip related features in the Alam et al. (2015) paper – e.g., their figure 8. I am not convinced of any of their visual 'signatures' of strike slip features and visual offsets.

Reply:

Thanks for this. As I said earlier their model lacks the basic architecture of a major dextral strike-slip fault system, thus any evidence they show is useless. Whatever they are showing is impossible, I am sorry for such strong words but it is true as per my understanding. When I say impossible I mean it. Structurally it is impossible to create a pull-apart basin the way they are showing it. That is the reason I have been very upfront about it. And I am convinced that such a structure does NOT exist at all. If someone can prove me the basic pull-apart model for Kashmir basin using the CKF structure then I might be wrong, and I will love to read such a discussion.

[Figure]

Fig. 1 a: After Shah, 2013

Fig. 1 b: After Schiffman, 2013

**Fig. 1.**

[Figure]

**Fig. 2 (after Shah, 2016)**

Shah, A., 2016, The Kashmir Basin fault and its influence on fluvial flooding in the Kashmir Basin, NW Himalaya, in Wessel, G.R., and Greenberg, J.K., eds., Geoscience for the Public 8 Good and Global Development: Toward a Sustainable Future: Geological Society of America 9 Special Paper 520, p. XXX–XXX, doi:10.1130/2016.2520(28).

**Fig. 2.**

---

## Referee Comment (RC2) · Anonymous Referee #2 · 21 Mar 2016

The manuscript is mostly concentrated in the confutation of what proposed by other authors on the tectonic origin of the Kashmir Basin (e.g. this manuscript bears the record of use of the word "impossible" among all the papers/manuscripts I read since now - 18 times, including figure captions in a 4 page manuscript). I do not intend to enter in the debate on its origin. As far as I understand, both interpretations may derive from not-sufficient, yet necessary evidences. And this strikes the right point. The aim of the manuscript is wrong following Popper: to prove a theory you require a sufficient evidence, yet you can get only necessary evidences that become sufficient only when you collect the total of them (that is, their number becomes infinite). On the other hand, one single evidence is sufficient to completely refuse a hypothesis. By considering the complexity of geological subjects (that Popper, for this reason, had difficulty in considering a science...) it is very easy to find exceptions on any proposed

theory/model/application. Due to this point, articles should always be propositive .The author is invited to propose, prove, and discuss an alternative model. In this way, both title and content do not arrive to any new/different conclusion and miss this point. Anyhow, in the following few observations on the weaker points of the manuscript are suggested. The title should be propositive. The confutation of the pull-apart model is forced to a very rigid interpretation and some possible indications (dashed lines) from the previous authors are interpreted as definitive sentences (e.g. Fig. 3). The confuted pull-apart is interpreted in a very rigid and surface way, without considering its extension at depth. I am sure that the author is well aware that pull-apart, as in the case of half-graben geometry, does not prosecute to the center of the Earth or until a ductile layer to absorb the displacement. Again, I am sure that the author knows well that a pull-apart structure most unlikely absorb/ re-equilibrate the entire displacement of a strike-slip fault. On the other hand, the generated structures/sediments should relatively migrate with the fault movement in the case of a complete displacement transfer. The presence of a limited zone of sediments is intrinsically an evidence of a dislocation along the basin borders. In a complex field as the Earth Science I would be very caution before stating "structurally impossible" without providing some quantitative evidence. The "piggyback basin" is used in the literature with slightly different situations and in this way it is generically applicable to many geological settings. In this case it was applied in 1982-83 and new evidences are required to update/confirm this interpretation. Its association with the attribute "classical" does not increase its reliability (as the "classical" Tolemaic model – of the sun orbiting around the Earth - with respect to Copernicus/Galilei ideas...) The pull-apart structure can be indifferently applied to left-lateral or right-lateral movements depending on the side you link to the strike-slip segments. A possible solution to the debate on the Kashmir Basin is that its origin might depend by mechanisms, an extension and a strike-slip component along a major fault due to a strain partitioning process, as often observed in other collisional chains (Alps). Many description looks like over-interpretations, e.g. "oval shaped basin" in page 1, line 27 (ref. to Fig. 1!). This is not so evident. Pag. 2 line 12 "structurally impossible" also

at depth? It is a 3D structure. Refer to previous comment on subsurface prosecution of the structure. Page 2, line 15 Warning: sinsedimentary faults do not necessarily have surface evidence, since they might be levelled by the youngest sediment deposition. The evidence is only when the displacement is faster than the sedimentation rate, and this might not be the case of Kashmir Basin, where continental sedimentation is produced by the faulting offset. Page 2, line 18 The author force the fault to be planar (I assume he intends also vertical!) and then describe the impossibility for a pull-apart to develop. You know that pull-apart can develop along transcurrent faults along zone of tilting of their dip and still maintaining their general strike. As mentioned, you can play by properly linking the strike-slip segments to the normal/transtensional faults and derive a right-lateral or a left-lateral sense of motion. Page 2, line 20 I agree with the author: interpretation of those minor faults as a horse-tail setting is difficult. Yet beware that horse-tail structures commonly consist of a set of normal faults. Page 3, line 14 Refer to the previous point on evidence from sin-depositional faults. In conclusion, the manuscript in its present form is not acceptable for publication. Hope that my suggestions will help in the preparation of a suitable manuscript.

---

## Editor Comment (EC1) · F. Rossetti (Editor) · 21 Mar 2016

Dear Dr. Shah,

I have not received two independent reviews of your ms. and both converge into the evaluation that your ms. is not suitable for publication. As noted by both reviewers (and I fully agree with their views), the ms. is intended to oppose previous interpretations presented in Alam et al. (2015) rather than to present independent, self-sustained data for alternative interpretation.

Actually, the submitted ms. appears as a "comment" paper rather than a standard scientific paper. On this subject, it is worth nothing you published on January 16 2016 a comment to the Geomorphology paper by Alam et al. (2015 -Tectonic evolution of Kashmir basin in northwest Himalayas; quoted with doi:

[Figure]

10.1016/j.geomorph.2015.03.025). Based on a comparative analysis of the two ms., overlapping is evident despite figures (but Fig. 1 is almost the same as the one in the comment) and text are somehow different. Arguments against the strike-slip hypothesis are, however, almost the same and therefore the submitted SE ms. duplicates what stated in the comment.

I therefore think your SE ms. does not add much to the scientific debate on the structural and tectonic setting of the region. Accordingly, my decision is to discourage the submission of a revised version and for ms. rejection at this stage.

Thank you for giving us the opportunity to consider your work.

Yours sincerely, federico rossetti

---

## Editor Comment (EC2) · F. Rossetti (Editor) · 21 Mar 2016

Dear Dr. Shah, I have now received two independent reviews of your ms. and both converge into the evaluation that your ms. is not suitable for publication. As noted by both reviewers (and I fully agree with their views), the ms. is intended to oppose previous interpretations presented in Alam et al. (2015) rather than to present independent, self-sustained data for alternative interpretation.

Actually, the submitted ms. appears as a "comment" paper rather than a standard scientific paper. On this subject, it is worth nothing you published on January 16 2016 a comment to the Geomorphology paper by Alam et al. (2015 -Tectonic evolution of Kashmir basin in northwest Himalayas; quoted with doi: 10.1016/j.geomorph.2015.03.025). Based on a comparative analysis of the two ms.,

overlapping is evident despite figures (but Fig. 1 is almost the same as the one in the comment) and text are somehow different. Arguments against the strike-slip hypothesis are, however, almost the same and therefore the submitted SE ms. duplicates what stated in the comment.

I therefore think your SE ms. does not add much to the scientific debate on the structural and tectonic setting of the region. Accordingly, my decision is to discourage the submission of a revised version and for ms. rejection at this stage.

Thank you for giving us the opportunity to consider your work.

Yours sincerely, federico rossetti

––––––––––––––––––––––––––––––